# Laterality in Horse Training: Psychological and Physical Balance and Coordination and Strength Rather Than Straightness

**DOI:** 10.3390/ani12081042

**Published:** 2022-04-16

**Authors:** Konstanze Krueger, Sophie Schwarz, Isabell Marr, Kate Farmer

**Affiliations:** 1Department Equine Economics, Faculty Agriculture, Economics and Management, Nuertingen-Geislingen University, Neckarsteige 6-10, 72622 Nuertingen, Germany; isy-marr@web.de; 2Zoology/Evolutionary Biology, University of Regensburg, Universitätsstraße 31, 93053 Regensburg, Germany; 3Behavioural Physiology of Farm Animals, University of Hohenheim, Garbenstr. 17, 70599 Hohenheim, Germany; sophie@schwarz-ffb.de; 4Centre for Social Learning & Cognitive Evolution, School of Psychology, University of St Andrews, St Andrews, Scotland KY16 9JPh, UK; katefarmer74@gmail.com

**Keywords:** balance, body asymmetry, equitation, horse, motor laterality, sensory laterality, stress, welfare

## Abstract

**Simple Summary:**

For centuries, straightening a horse has been considered a key element in achieving its responsiveness and suppleness and has been a traditional goal in training. However, body asymmetry (natural crookedness), motor laterality (the preference for limbs on one side) and sensory laterality (the preference for sensory organs on one side) are naturally occurring phenomena. In humans, the forced correction of these imbalances, for example, forcing left-handed children to write with their right hands, has been shown to lead to psychological imbalance. In view of this, lateral asymmetries in horses should be accepted, and training should focus on psychological and physical balance, coordination and equal strength on both sides, instead of enforcing “straightness”. To explore this, we conducted a review of the literature on motor and sensory laterality in horses and found that the evidence suggests that enforcing straightness may be stressful and may even be counterproductive by causing psychological and physical imbalance relative to a horse, making it tense and uncooperative. In general, body asymmetry has been shown to have little impact on performance, but increases in motor and sensory laterality can indicate insufficiencies in housing, handling and training. We, therefore, propose that laterality should be recognized as a welfare indicator and that straightness in a horse should be achieved by conducting training focused on balance, coordination and equal strength on both sides.

**Abstract:**

For centuries, a goal of training in many equestrian disciplines has been to straighten the horse, which is considered a key element in achieving its responsiveness and suppleness. However, laterality is a naturally occurring phenomenon in horses and encompasses body asymmetry, motor laterality and sensory laterality. Furthermore, forcibly counterbalancing motor laterality has been considered a cause of psychological imbalance in humans. Perhaps asymmetry and laterality should rather be accepted, with a focus on training psychological and physical balance, coordination and equal strength on both sides instead of enforcing “straightness”. To explore this, we conducted a review of the literature on the function and causes of motor and sensory laterality in horses, especially in horses when trained on the ground or under a rider. The literature reveals that body asymmetry is innate but does not prevent the horse from performing at a high level under a rider. Motor laterality is equally distributed in feral horses, while in domestic horses, age, breed, training and carrying a rider may cause left leg preferences. Most horses initially observe novel persons and potentially threatening objects or situations with their left sensory organs. Pronounced preferences for the use of left sensory organs or limbs indicate that the horse is experiencing increased emotionality or stress, and long-term insufficiencies in welfare, housing or training may result in left shifts in motor and sensory laterality and pessimistic mentalities. Therefore, increasing laterality can be regarded as an indicator for insufficiencies in housing, handling and training. We propose that laterality be recognized as a welfare indicator and that straightening the horse should be achieved by conducting training focused on balance, coordination and equal strength on both sides.

## 1. Introduction

It has long been known that horses have lateral biases, and they are often described as having a “hollow side” and a “stiff side”. For centuries, a goal of training in many equestrian disciplines has been to straighten the horse [1,2], and the frequently cited “scale of training” considers straightening a horse a key element in achieving responsiveness and suppleness. Physical one-sidedness is considered an undesirable trait [1,2] because it may impact the horse’s gaits [2,3], performance [4,5] and orthopedic welfare [6,7].

However, laterality is a naturally occurring phenomenon in horses, as in other mammals [8] and in some invertebrates [9], and encompasses body asymmetry, motor laterality and sensory laterality. Body asymmetry is defined as the unequal anatomy of the left and right side of the body and manifests in a crookedness of the longitudinal axis of the body [2,10]. Motor laterality describes the preference of an animal to use the limbs on one particular side [11], and sensory laterality describes a preference for the use of sensory organs, such as eyes and ears, on one side [12]. Body asymmetry is innate, while motor and sensory literalities are partially innate and are partially formed through functions and information processing of the asymmetrical brain hemispheres [8].

In humans, forcibly counterbalancing motor laterality, e.g., forcing left-handed children to write with their right hands, has been considered a cause of psychological imbalance [13,14,15]. Therefore, the call for “straightening“ the horse may have been misunderstood or may even not be appropriate. Perhaps asymmetry and laterality should rather be accepted, with a focus on training psychological and physical balance, coordination and equal strength on both sides instead of enforcing “straightness”.

To explore this, we conducted a review of the literature on the function and causes of motor and sensory laterality in horses, especially in horses when trained on the ground or under a rider. As others have questioned the biomechanical implications of “correcting” body asymmetry and motor laterality before [2,10], this review will not go into detail on this aspect but will focus on the impact of “straightening the horse” on the psychological welfare of horses and the meaning of motor and sensory laterality. We will describe body asymmetry only briefly when its effect intersects with motor and sensory laterality [2,10]. In the final discussion, we will debate options for “straightening the horse” and the benefit of using motor and sensory laterality as indicators for animal welfare in horse training and management [16].

## 2. Forms and Measurements of Laterality in Horses

### 2.1. Body Asymmetry in the Horse

Body asymmetry in horses is complex and may have many different forms [2,17]. Generally, it is defined as the unequal anatomy of the left and right side of the body. It manifests itself in a crookedness of the longitudinal axis of the body, in a curve to the right, to the left or in an S-form. When moving, some horses displace their shoulder or their hindquarters to the left or right, which results in sideward movements of different degrees [2]. Furthermore, horses may drop their withers on one side [10] or move their hindlegs unequally [2,18,19], without showing any indications of pain. Probably as a result of such complexities, riders’ reports on their horses’ body asymmetry often do not match measurements taken with electric markers attached to the horse’s body when walking and trotting on a treadmill [2].

### 2.2. Motor Laterality in Horses

Motor laterality is common in horses and can be divided into foreleg preference laterality, sometimes addressed as handedness, and hindleg preference laterality, also addressed as footedness [2]. Studies on ridden horses have shown significant one-sided use of the preferred limbs [20,21]. A common method of assessing handedness in horses is observing the foreleg position while grazing, the grazing stance [20], as many horses prefer to graze with one particular foreleg in front of the other. Other studies have found a preference for a certain foreleg when starting to walk [22,23]. Footedness can be evaluated by observing the direction of flight responses and observing which hindleg bears most body weight when the horse turns [18].

### 2.3. Sensory Laterality in Horses

Sensory laterality is expressed through the one-sided use of sensory organs and can be assessed by observing, for example, the ears and eyes favoured when the horse is presented with novel objects or humans [24]. Several studies have focused on visual laterality [18,25,26] in animals that have to shift their heads clearly to one side for information intake, as the lateral positioning of the eyes makes it easy to observe [16,27,28,29,30]. Left sided sensory organ use has been found to be particularly strong when the stimulus is connected with an emotion, such as fear [28].

## 3. Causes and Functions of Laterality

### 3.1. Body Asymmetry Is Innate

Body asymmetry has been observed soon after birth in 80–90% of humans [31], toads [32,33], chickens [34] and horses [35], and is said to be caused by the unequal distribution of organs in the body and the position of the embryo in the womb during the gestation period [35]. Therefore, it can be termed innate and morphologic.

### 3.2. The Lateral Brain Forms Motor and Sensory Laterality

Motor and sensory laterality are partially innate (i.e., morphologic) and partially formed by asymmetrical functions and information processing of the brain hemispheres (i.e., cerebral). In a number of species, the left brain hemisphere is responsible for well established, routine behaviour, often based on learned patterns, such as finding food, and most animals use the left hemisphere to control proactive behaviour and specific subjects or tasks [36]. As a result, the left hemisphere is mostly dominant in situations that are well known to the animal and not perceived as stressful or dangerous [37]. Therefore, it is not surprising that there is a correlation between the use of the left hemisphere and a positive cognitive bias, i.e., positivism, in many animals [22,38]. The right hemisphere is connected with a negative cognitive bias, i.e., negativism, and with reactive behaviour [36]. Therefore, the right hemisphere appears to control strong emotional arousal and global attention, such as looking out for predators. It dominates in stressful or even life-threatening situations and controls fast responses, such as fight or flight. The right hemisphere also controls the physiological parameters of stress, such as the release of stress hormones, and the heart rate (summarised by [36]).

The left and right hemispheres are neurologically connected, mostly to the contralateral side of the body. Therefore, information received through the right eye, ear and tactile senses will largely be processed in the left hemisphere, which mostly controls the movements of the contralateral, i.e., the right, limbs [39], while information perceived by the left eye, ear and tactile senses is processed in the right hemisphere and given back to the left limbs [39]. However, information gathered by the nostrils is processed by the ipsilateral hemisphere, i.e., the same side of the brain [8,12,36]. This is, presumably, because the two rostral nasal cavities open into one communal cavity in front of the ethmoid bone, where olfactory nerves pass into the brain. Therefore, incoming olfactory information from the left and the right nostrils mix in the nasal cavity and are not clearly separated before they are transferred to the brain [40].

### 3.3. Laterality-An Evolutionary Advantage

Some studies propose that laterality increases the efficiency of the brain and, therefore, represents an evolutionary advantage [41]. Sensory and motor literalities are said to be beneficial for the survival of animals under threat. For example, studies on recently hatched chicks showed that those with stronger sensory lateralization were less likely to be caught by predators, as they could simultaneously eat and be on the look-out [42]. Rogers [43]; therefore, this suggests that the strength of laterality might be more important than its direction.

Another example of the evolutionary advantage of laterality in a prey animal is in the horse [37]. For horses to survive in the wild, it is crucial for them to be attentive to their surroundings and be able to react to possible threats, while simultaneously performing other tasks such as seeking food. For example, feral horses [37] increasingly observed the environment with their left sensory organs when agonistic interactions arose within the group. The greater the level of aggression and the need for fast reaction, the stronger the need to process this information in the right brain hemisphere and to keep the left brain hemisphere free for rational decision making.

Therefore, the right brain hemisphere appears to be in charge of processing agonistic interactions as well as potentially threatening situations [37]. Similar results have been observed in other animals in other studies, and this, therefore, supports the theory that the right hemisphere in vertebrates is dominant in stressful and unpredictable situations ([36], see further below), which eventually provides an evolutionary advantage in avoiding danger.

## 4. Sensory and Motor Laterality in Horses

### 4.1. Laterality in Feral Horses

Australian feral horses have been found to show left-bias in sensory laterality in the form of head posture and eye preference during agonistic interactions but, interestingly, there was no significant motor laterality on the population level [37]. About 40% demonstrated a left leg preference while grazing, and about 40% demonstrated a right leg preference, while about 20% were ambidextrous, with young horses being more strongly lateralised than mature horses [37]. The study was later repeated on Przewalski horses [44] and, again, no significant population bias in motor laterality in terms of forelimb preferences was found. About 30% were left leg biased, 30% were right leg biased, and 40% were bilateral [44]. Compared with domestic horses, both Australian feral horses and Przewalski horses showed a stronger left-lateralisation in eye preference and head posture in aggressive and high alert behaviour [44]. These three studies confirmed that lateralization plays an important role in responses to potential threat and in social interactions [44].

### 4.2. Sensory Laterality in Social Interactions between Horses

The proposed association between right hemisphere use in horses and negative reactivity was further evaluated in the context of social interactions [45]. Following the finding of feral horses preferentially using the left sensory organs in agonistic encounters [37], Farmer et al. [45] assessed sensory organ use in affiliative interactions, such as social grooming, between group members. Social groups of ponies and riding horses showed significant left-eye preference in affiliative interactions with conspecifics [45]. There was a tendency for the riding horses to show stronger laterality than ponies, possibly supporting the theory of differences in laterality on a breed and training level [21,25]. As sensory lateralisation was found to be independent of age, sex and rank of the test horses, it was concluded that the right brain hemisphere might be responsible for processing social situations in general. Therefore, Farmer et al. [45] suggested that the level of attention and emotionality seemed to be more important in influencing which hemisphere dominates in the given situation than whether horses experience positive or negative emotions. This hypothesis is strongly supported by findings in a number of vertebrate species, suggesting that the strength of the laterality is often more relevant than its direction [43].

In line with a general left-eye preference in social interactions, Karenina et al. [46] found evidence that a left-side bias is already present in interactions between mares and their foals, as in many other social mammals. The preference amongst infants of various species to keep their mothers on their left, i.e., in their left sensory fields, may well be due to processing social and emotional information in the right brain hemisphere.

### 4.3. Auditory Sensory Laterality in Communication between Horses

Along with horses’ sensitivity to sound and ability to direct each of their ears independently towards a sound source, studies have shown that horses also express auditory laterality [47]. A group of horses was exposed to recorded whinnies of other horses, and they showed auditory laterality depending on the social implications of the calls. When the calls came from stable neighbours, but not from group members, the horses had a significant right ear preference, while calls from group members or strangers did not cause significant lateral responses. However, there was a trend for increased left-ear use as a reaction to unfamiliar calls. Basile et al. [47] concluded that since there was no clear difference in lateralised responses between familiar and unfamiliar calls, it is likely that the level of emotionality aroused by the stimulus strongly influences laterality. It is possible that the calls of familiar non-group members (stable neighbours) cause a higher state of attention and horses do not consider unfamiliar calls to be meaningful.

### 4.4. Sensory Laterality Reflects Emotional Responses When Horses Are Confronted with Objects

In many studies, sensory laterality was analysed by testing the horses’ responses to novel objects. For example, Larose et al. [25] tested sixty-five horses of two different breeds, Trotters and French Saddlebreds, for their visual laterality in a novel object test. They found a significant correlation between the emotional behaviour of the horse and the direction of its visual laterality: the higher the emotionality index, the higher the probability that the horse would investigate the object with its left eye. Interestingly, this study revealed a difference in visual laterality between the breeds, with higher emotionality, and corresponding laterality, in Trotters. This could have been due to different usages and training of the breeds, or to genetics, with lower emotionality in French Saddlebreds. The general correlation between a preference of the left eye for novel objects and high emotionality is reflected in a number of studies in other species as well [8].

However, in another study, De Boyer Des Roches et al. [24] compared visual and olfactory laterality with respect to the emotional value of objects in Arabian horses. The preferred side of sensory approaches and the difference between monocular and binocular visual usage were examined in relation to a positive object, a negative object and a neutral, novel object. The neutral, novel object was mainly investigated with the right eye, there was a tendency to observe the negative object with the left eye, and no lateral bias was observed for the positive object. This supports the theory that novelty and emotionality increase sensory laterality but do not necessarily predict the direction [24].

In addition, sensory laterality with respect to novel objects has been shown to change with training. Marr et al. [48] observed left-eye preference in horses trained on the left, and when the training was changed to bilateral methods, there was a significant right-shift of the horses’ visual laterality. Furthermore, Marr et al. [48] assumed a connection between the horse’s laterality and certain character traits and called for further investigations on this subject.

Considering the correlation between left-eye preference and emotionality, Austin and Rogers [18] examined whether horse’s reactivity to a novel object would change depending on which side it was presented. They measured horses’ speed and distance of retreat from a person approaching while opening an umbrella. The horses that were approached from the left showed significantly stronger flight reactions than those approached from the right. The procedure was then repeated for each horse from both sides in a random order, and it was found that left-side-approach reaction was stronger in the horses that had not already been approached from the right side. Therefore, Austin und Rogers [18] suggested that there was a transfer of information between the hemispheres causing the right hemisphere in the second experiment to register the umbrella as an already known and not implicitly dangerous object.

### 4.5. Body Asymmetry Affects Motor Laterality

Immediately after birth, foals’ body asymmetry may already affect their predispositions for motor laterality in their grazing stance, i.e., the placement of one foot in front of the other while grazing. Van Heel et al. [6] showed that foals that express body asymmetry and motor laterality at a young age had a higher chance of developing uneven feet (that is to say a greater than 1.5° difference between the angles to the ground of the inner and outer hoof walls). This may continue to affect the foal’s forelimb preference and may have a significant impact on the horse’s physiology and later performance. In a follow-up study with the same horses, van Heel et al. [6] found that the correlation between motor laterality and unevenness of the feet increased in maturing horses. Horses with little unevenness in the feet displayed only weak motor laterality, while those with strong unevenness in the feet showed strong motor laterality, which was assumed to have a negative impact on sporting performance [7].

### 4.6. Motor Laterality in Trained Horses

From two studies with equal numbers of left and right lateralised feral horses, Austin and Rogers concluded that unique forelimb preferences on a population level may result mostly from domestication (i.e., breeding) and training [37]. McGreevy und Rogers [20] showed that Thoroughbred horses prefer to position the left foreleg in front of the right while grazing. This expression of motor laterality increased with the age and duration of training of the observed Thoroughbred horses [20], while Quarter Horses showed a stable distribution of equal numbers of left and right motor lateralised animals [21]. As Thoroughbreds trained on an oval training course in clockwise and counterclockwise directions displayed a stronger left front leg preference in their grazing stance, it has been suggested that the difference between the breeds is due mostly to differences in emotionality, with Thoroughbreds being generally more emotional than Quarter Horses [21].

Lucidi et al. [35] suggested that the development of motor laterality in trained horses may be revealed by the direction in which horses reduce the diameter of a circle and come closer to a person in the middle of the circle when being lunged, termed “cutting a circle”. They found that bilaterally trained horses, with increasing age, increasingly cut the circle when lunged on the right hand. Only nine of the twenty-nine nine-month-old foals (39%) were observed, but ten of the seventeen two-year-old horses (59%) cut the circle. The increased cutting of the circle in older horses led to the conclusion that this derailment may not primarily be caused by body asymmetry but may be caused by the development of motor laterality. They also concluded that typical mounting and handling from the left side are probably not the only causes of the development of motor laterality, as that would be more likely to cause left-sided motor laterality in horses when balancing the weight of the person when they were mounting from the left side [35].

In another experiment, fifteen unridden horses and fifteen horses that had already been ridden were tested for motor bias by Wells and Blache [49]. They compared two different test conditions. The first was lunging in canter on a circle, which actively challenged the balance of the horses. The second was forelimb preference while grazing, which did not require active balancing from the horses. Interestingly, they found no difference in laterality between the unridden and ridden horses during lunging, with neither groups showing a motor bias in canter. There was, however, a significant difference in grazing observations. While the younger, unridden horses again showed no signs of motor bias, the ridden horses had a right-forelimb preference. Therefore, Wells and Blache [49] concluded that even though it was not expressed when the horses had to actively balance themselves, they seemed to have acquired motor laterality with increased training and/or age, supporting the observations made by McGreevy and Rogers [20]. An important factor influencing the laterality of the ridden horses was that the tested horses were usually ridden and trained equally from right and left sides, which may have resulted in weaker lateralisation [49].

All these findings together suggest that motor laterality is modulated by the horses’ age [20], breed [21] and training [50]. Kuhnke [17] found left-biased horses to be more successful than right-biased horse in dressage and jumping disciplines. In Thoroughbred and Arabian racehorses that gallop around an oval racetrack in an anticlockwise direction and in Quarter horses that race on a straight line, the lateralisation of stride patterns has also been linked to performance [51], with a more advantageous ratio between strides and the intake of oxygen when the horses ran with a right-sided stride pattern. Switching the leading leg when galloping around corners seemed to be advantageous when Thoroughbreds were running on an oval training course in both clockwise and counterclockwise directions [21].

### 4.7. Sensory Laterality in Horses in Contact with Humans

Farmer et al. [26] investigated whether sensory laterality can be observed between horses and humans. Creating an experimental setup in which the horses had to pass a person to reach a food source, the horses had the free choice of whether to pass on the left or the right. To exclude the often-discussed influence of one-sided handling and training of the horse causing laterality [21,37,44,52], the experiment was repeated with bilaterally trained horses. Furthermore, the horses were tested both with familiar and unfamiliar persons as an “obstacle” they had to pass, as well as in interactive situations in which the horse approached a familiar person. In all cases, the horses expressed left-eye laterality, which was stronger in left trained horses and in interactive situations. The fact that both the bilateral and the left trained groups were left biased implies that training was not the only cause of sensory laterality. Farmer et al. [26] concluded that the left eye and right hemisphere were preferred in situations that required the initial evaluation of the quality of information, such as the presence of a human. These findings are consistent with Farmer et al. [45] showing left laterality in affiliative interactions, suggesting that not only negative emotions but, at least to some extent, also positive emotions are processed in the right hemisphere, as discussed by Killgore und Yurgelun-Todd [53].

Furthermore, it has been suggested that, in horses, the left brain hemisphere is responsible for processing visual and auditory information received from familiar individuals of their own species and of other species, such as humans [54]. On an inter-species level, horses have been able to match a familiar person correctly to the recording of that person’s voice. Proops and McComb [54] played a voice recording directly in front of each horse, with two people standing on each side of the sound source, in the horse’s monocular fields of vision, and the horses correctly matched the person with the voice when the person was standing on the horse’s right side. Therefore, Proops und McComb [54] concluded that the familiar visual and auditory stimuli were combined more successfully in the left hemisphere.

However, all horses in the Farmer et al. [26] and Proops and McComb [54] experiments were used to being handled by humans, and another study by Sankey et al. [55] compared the reaction of untrained one-year-old horses when approached by humans with the reactions of already handled two-year-olds. Interestingly, Sankey et al. [55] found that the untrained horses showed a significantly higher number of negative reactions, such as avoidance, and threats to kicks and/or bite towards a person approaching the horses in its the left monocular field. The two-year old horses, on the other hand, did not display any correlation between negative behaviour and the side of approach [55].

Left sensory organ use in contact with humans may also increase when the persons are considered a threat. In a study by Smith et al. [56], horses preferentially approached pictures of persons with the left eye in all situations, but this was expressed more strongly when the pictures showed a person with an aggressive facial expression. The same appears to be the case for auditory laterality, as horses reacted to voices of persons they matched to negative prior experiences predominantly with the left ear [57].

These findings reinforce former study results on two levels. Firstly, they strongly support the correlation between laterality and a horse’s age and training, indicating that the horse’s experience is an important factor. Furthermore, the negative behavioural reactions shown in combination with human approaches from the left, the strong preferences for left sensory organ use for “unpleasant” persons and a corresponding activation of EEG wave patterns in the right brain hemisphere [57] are consistent with the theory that the right brain hemisphere is responsible for negatively connotated reaction [55]. Finally, in this study, horses that were approached from the right side displayed positive behaviour as a reaction, such as immobility and turning the head to sniff the approaching person in contrast to threatening the persons which approached from the left [55] and, therefore, indicated left hemisphere processing of neutral or positive emotions [24,58].

### 4.8. Body Asymmetry, Motor and Sensory Laterality under a Rider

Although equitation is one of the main components of horse domestication, so far, there have been very few studies investigating whether a rider has a direct influence on the horse’s body asymmetry and motor laterality. Body asymmetry under the rider was measured at the level of the horses’ withers, and 60% of the horses were found to be larger on the left than the right side of their thorax [59]. Thoracic asymmetry was not affected by the horses’ breed, age, sex, height or level of training or the riders’ age, gender, height, weight and level of training [59]. However, Cocq et al. [60] found increased lateral body asymmetry in terms of an increased lateral bending of the back, when the horse was ridden compared with unridden. Furthermore, Kuhnke [17] investigated whether objective measurements of motor laterality were consistent with the horse’s body asymmetry as perceived by the rider but concluded that most methods were not suitable for measuring body asymmetry and motor laterality in horses under the rider. This was consistent with the findings of Rehren [2] who found that body asymmetry assessments by riders were not consistent with body asymmetry measurements on a treadmill.

Murphy et al. [52] found the motor laterality of male horses to be similar whether ridden or not on an obstacle avoidance test, while female horses displayed a greater degree of motor laterality when ridden. A recent study on the impact of a rider, who provided minimal aids, on motor and sensory laterality in male and female riding horses found that, in both sexes, motor laterality increased when stepping over a pole while carrying a rider [61]. However, in sensory laterality tests, there was no significant effect on the rider [61]. Interestingly, the ridden horses in this study showed no significant population level left or right bias in motor laterality, similarly to feral [37] and Przewalski horses [37].

Additionally, Kuhnke et al. [62] showed that rein tension while riding is influenced by the motor laterality of the horses as well as by the laterality of the rider, i.e., their handedness. The eleven test riders were all right handed. The horses’ motor laterality was assessed by the owners and confirmed by a foreleg preference test, modified from van Heel et al. [6], where a foreleg preference was measured when horses stopped in front of a bucket after 20 approaches from varying distances. The results for the rein tension showed that the riders applied different strengths and ranges of tension on the different reins depending on whether the horse was left or right lateralized. The differences in rein tensions were consistent with the riders’ conscious or unconscious efforts to straighten the horse, so the laterality of horse and rider may have an impact on training that is rarely considered.

### 4.9. Increased Laterality Is Necessary When Horses Are Stressed

Stress-induced changes in motor laterality, for example, the preference to use a particular forelimb, have been observed in mice [63], lions [64] and donkeys [65]. Stressful situations have also been shown to have a significant effect on lateralisation in horses [16,66,67]. Siniscalchi et al. [66] found a left-side preference in forelimb use when horses were confronted with stressful tasks, such as loading for transport, which induced increased anxiety behaviours. Furthermore, Siniscalchi et al. [68] observed a correlation between the strength of sensory (i.e., right nostril), laterality and increased heart rate in horses trained for jumping. As information from the right nostril is processed by the right brain hemisphere, this supports the theory that the right brain hemisphere is responsible for an increased lateral bias in stressful situations.

When comparing laterality with a stress-associated parameter such as stress hormones (glucocorticoid metabolites, GCMs), horses with increased levels of faecal stress hormones expressed increased left-side preferences in both motor and sensory laterality [67]. Additionally, Marr et al. [16] investigated the difference in motor and sensory laterality in response to stressors. Moreover, by using faecal GCM as an indicator for stress, they examined the effect of initial training and a change of housing on young horses. The social stressor of being changed from group housing to individual housing caused immediate increases in GCM levels as well as a significant left shift in sensory laterality [16]. The GCM levels then declined slightly but remained elevated from baseline levels throughout the experiment. Motor laterality, on the other hand, took longer to change, but the shift was measurable after one week of the new housing situation. Marr et al. [16] found that the significant left-shift in motor laterality, such as GCM levels, lasted for the duration of the experiment and, therefore, could indicate long-term stress. Sensory laterality, however, changed more quickly and was more situation related, and the authors suggest that sensory laterality may be a helpful parameter in detecting acute stress.

In horses, the left-sensory organ use bias in correlation with stress seems to be consistent on a population-level [16,66]. Furthermore, recent studies that applied EEG wave measurements to brain activations support the finding that the right brain hemisphere is activated under stress [57,69], and along with this brain wave activation, the horses also display left sensory organ use [57], as well as elevated stereotypic behaviour [69], which has long been considered to be a strong indicator of reduced welfare [70]. Therefore, it has been suggested that laterality could be a valuable indicator for welfare evaluations.

### 4.10. Manifested Laterality Reflects the Mentality

In view of the association between laterality and stress, it is not surprising that laterality has been considered a possible indicator for an animal’s mental state [22,71,72]. Motor laterality and cognitive bias for optimistic or pessimistic information processing have already been linked in other species, such as common marmosets [38]. Marr et al. [22] observed a significant correlation between forelimb preference and cognitive bias in horses. Optimistic horses that judged a neutral stimulus to be positive were more likely to use the right forelimb first when they started moving, while pessimistic horses preferentially started to move with the left forelimb.

Furthermore, laterality reflects the degree of emotionality horses experience in training [16], and the degree of their emotionality in return impacts their training. High emotionality has been shown to negatively affect trainability [73]. It would, therefore, seem appropriate to take motor and sensory laterality into consideration as indicators for increased stress [18]. Laterality also has safety implications, as strongly lateralised horses may show strong reactions to a stimulus, which could result in flight and endanger the animal as well as the rider or handler [20,74].

### 4.11. Training and Housing of Horses Should Consider Motor and Sensory Laterality

A left shift in motor laterality during training could be used as an indication that the training is stressful to the animal [16,48]. D’Ingeo et al. [57] found that horses responded with their left ear to human voices they associated with negative prior experiences and with their right ear to humans they connected to positive experiences [57]. The preference for left ear responses increased when horses were kept under restricted conditions, such as individual box housing with limited pastures [57].

Other studies have reported that horses experience increased right brain hemisphere activation when kept under conditions that compromised basic needs [57]. As mentioned before, stress from a change from social to individual housing and the commencement of initial training caused a significant left-shift in motor and sensory laterality [16]. Horses under conditions of restricted basic needs, such as limited free movement and social contact [70], in combination with use as riding school horses, displayed right brain EEG wave pattern activations and increased displays of stereotypic behaviours [69], which are a strong indicators that the animals were suffering from poor environmental conditions [70]. A reduction in available space has been shown to cause a shift in the motor laterality of donkeys [65]. The donkeys initially showed a right-side motor bias on the population level, which diminished as their space was reduced and the animals started to use their left limbs more often. All the above indicate that a left shift in laterality appears to be a good indicator of reduced welfare in equine training and husbandry.

Rogers [36] proposed that the asymmetrical functions of the brain hemispheres should be observed closely to reduce the impact of training on welfare. Austin und Rogers [18] suggest that an animal should ideally be exposed to a novel stimulus from the right side, because triggering the stress-related right brain hemisphere should be avoided. However, other studies indicate that many horses prefer to use left sensory organs to investigate novel stimuli [16,22,26,45,46,47,54], presumably because information processing in the corresponding right brain hemisphere is faster [25], which may yield survival benefits in cases of potential danger [42]. Marr et al. [48] showed that bilateral training, i.e., handling the horses equally from the left and the right, could reduce strong left-sided visual laterality. This may imply that alternating training between the left and the right could potentially lead to decreased stress levels in training and other situations.

## 5. Discussion

The following sections will consider whether the traditional training goal of straightening the horse [1,2] is appropriate and discuss whether it is beneficial for the psychological and physiological welfare of the horse to change morphological body asymmetries or cerebral motor laterality and sensory laterality.

### 5.1. Is Body Asymmetry Maladaptive?

In ridden horses, body asymmetry has been said to cause difficulties in balancing the body and the rider and to affect movement patterns and anatomical structures. Racehorses with severe body asymmetry were shown to have below-average performance, and those with strong motor laterality of the forelimbs reached racing qualification late [75]. Furthermore, endurance horses with strong body asymmetries suffer orthopaedic complications [5] more frequently. Horses with strong body asymmetries were shown to bear more weight on the left front leg while moving their shoulders to the right when trotting [54] and, therefore, demonstrate stronger motor laterality. Some studies have proposed that orthopaedic issues arise from strong biases in body asymmetry, which manifest themselves in motor laterality, such as in a consistent preference for placing one leg in front in the grazing stance [6,7]. A strong preference for one front leg while grazing has been said to cause unevenness in the hoofs and uneven forces on the tendons and lower joints of the legs, and this may, eventually, result in poorer performance [6,7]. Finally, body asymmetry and motor laterality in the hindquarters have been said to affect the movement of the horse’s back [3] and are assumed to result in orthopaedic issues [19].

However, it remains debatable whether the studies above confuse the cause and effect of strong body asymmetry and poor performance and health issues, as from the genetic perspective, 80% of horses are born with a body asymmetry and 60% show thoracic asymmetry under a rider [59]. Regardless of this, most are likely to be capable of high physical performance, and their physical structures such as bones, muscles, ligaments and tendons adapt to crookedness [76]. Horses with significant body and movement asymmetries do not display a higher rate of lameness [10]. Furthermore, orthopaedic research suggests that it is more advantageous to accept crookedness in the body, since the morphology and the body structures have already adapted to it. Correcting such physical crookedness may even cause stress on bones, muscles, ligaments and tendons and, consequently, may damage them [10,77].

### 5.2. Is It Beneficial to Change Motor Laterality through Training?

While morphological body asymmetry is understood to be innate and will, consequently, be difficult to manipulate, motor laterality has been shown to be partly innate and to develop with age in maturing horses [20] and is partly acquired through development of the brain through controllable factors such as training [49] and stress [16]. However, innate and acquired motor laterality show a strong tendency to manifest themselves in maturing individuals within the first years of life and are, therefore, connected to the cerebral functions. Consequently, enforced changes in motor laterality are no longer practiced in humans and have even been implicated in psychological imbalance [13,14,15]. Human handedness is now considered important for correctly processing information in the brain. Rehren [2] suggested that attempts to erase a horse’s motor laterality could have a similar effect. Forcing a horse into movements by trying to change its laterality could cause stress and can activate the right hemisphere, which is responsible for emergency responses. Due to the cerebral connection with laterality, the overactivated right hemisphere could then even increase left-sidedness. It may finally result in a negative cognitive bias (i.e., pessimism) along with a manifested left motor bias [22,71].

In addition, as with the horses’ body asymmetry, strong motor laterality does not necessarily prevent them from being successful in sport, and it may even be beneficial for their performance [17,51]. Some studies have linked a certain direction of motor laterality to higher performance, such as right-lateralised racehorses being more successful in races [51].

### 5.3. Is It Beneficial to Change Sensory Laterality through Training?

Strong sensory laterality provides an advantage in survival and performance in a variety of animals. [42,43]. Since one-sided information intake is transferred directly to the corresponding brain hemispheres, animals are able to choose the relevant side of sensory organs to evaluate the environment, situation or other animals.

The fact that sensory laterality plays an important role in social interactions in horses supports this theory. Many horses prefer to keep conspecifics and humans in their left visual field because social interactions are thought to be processed in the right hemisphere [45]. In flight animals, such as horses, this has the effect in which multiple individuals in a herd can flee quickly and in a coordinated way in cases of a threat, and this side preference already plays a role in mother–foal interaction [46]. In short, sensory laterality in horses is a deeply entrenched trait to ensure survival. In other species, strong laterality in individuals has also been connected with better chances of survival and has, therefore. been suggested to have an evolutionary advantage, such as coordinating cognitive decision processes and fast fight or flight responses [41]. Furthermore, greater sensory laterality has been said to provide advantages in the form of better cognitive performances [43].

It has been shown that the response to external stimuli can differ greatly depending on the eye the horse used for information intake [18]. They may react fearfully when information is received on the left side and less strongly when information is received from the right side [18]. However, the suggestion that it may be better to approach and present objects from the right in order to avoid strong reactions [18] again may confuse causes and effects. Strong reactions of horses to human approaches or objects may originate from stress due to previous experiences in human training [57,69] and/or from unsatisfactory management conditions [16]. Because stress increases right brain hemisphere activation [57,69], horses need to use left sensory organs for information input. Consequently, especially for species such as the horse, with laterally placed eyes and a large area of monocular vision [78], the freedom to turn their heads to assess their environment is crucial. Furthermore, it is to be assumed (Krueger, unpublished data) that preventing horses from using a preferred sensory organ by turning their heads or being led on a certain side may disrupt the correct processing of information in the relevant hemisphere and, therefore, lead to misevaluation and errors in their decision making and, therefore, stress.

### 5.4. Moderate Sensory Laterality May Reflect Trust in Humans

Horses that trust humans in training and handling may experience reduced emotionality and display reduced sensory laterality. We may conclude this from the following facts: (a) A preference for left sensory organ use has been shown to be higher in contact with unknown persons [26]; (b) left sensory organ use increases when horses are confronted with objects or situations which cause negative emotions [24,25]; and, finally, (c) sensory laterality gradually decreases with bilateral training [48].

### 5.5. Laterality as An Indicator for Stress

Laterality can be a valuable indicator in assessing the horse’s mental state [22,72] (i.e., optimism versus pessimism) and stress level [16]. Stress has been shown to be common in equestrian sport and can have a multitude of different causes [79], and the consequences of stress and reduced welfare can even cause long-lasting depression [80] and pessimism [22,71,72]. In the past, stress to a horse under a saddle has been measured via different behavioural and physiological parameters [79] such as heart rate; the cortisol level in saliva or blood; and cortisol metabolites in the horse’s faeces [79]. Laterality has consistently been shown to be connected with stress [16,66,67], and recent EEG measurements confirmed increased right brain hemisphere activation in stressed horses [57,69]. Carrying a familiar rider using minimal aids did not increase sensory laterality in experienced riding horses [61], and changes in laterality, especially to the left, could be considered an additional parameter for evaluating the impact of training, management and handling on the horses’ mental state and stress level.

### 5.6. Benefits of Balance, Coordination and Equal Strength

Just as in human athletes, a sport horse also has to be able to use both sides of its body, even when these sides are not symmetrical. However, it remains debatable whether many riders confuse straightening the horse by reducing body asymmetry with increased straightness through counterbalancing, because body asymmetry and motor laterality assessments by riders have been shown to mismatch measurements taken from the ground [17]. Under the weight of a rider, horses’ motor laterality [61] increases to its preferred side, either to the left or to the right [52,61], and the harmonious use of both sides is believed to be a prerequisite for success in sport and the long-term maintenance of the horse’s health. To achieve this, the horse needs appropriate training to build equal strength on both sides, as suggested by many trainers.

Furthermore, true “balance” is not only lateral; it is also longitudinal: The horse has to be balanced between hindquarters and forehand, as well as between left and right, and often “straightening” only becomes an issue when riders and trainers try to deal with the longitudinal balance too soon or too severely [1]. If the horse’s head and neck are pulled back and downward to bring the forehand back over the balance point, rather than bringing the hindquarters forward and under the horse’s balance point, horses may displace laterally to “escape”, because the hindquarters are not yet strong enough to carry the horse in true balance. This calls for using a variety of exercises to strengthen the hindquarters, as well as bilateral exercises to strengthen each side of the horse, in order to help it find its balance, while accepting its natural laterality [48].

## 6. Conclusions

In view of recent research, the goal of straightening the horse should perhaps be reconsidered [10]. Body asymmetry is innate, but it does not prevent the horse from performing at high level under a rider. Many methods proposed to achieve straightness, such as additional equipment and forced training on the weaker side, may be stressful to the horse [1,77] and may even be counterproductive by causing the horse to become tense and uncooperative [2,10]. In the worst case scenario, this can lead to a loss of sensitivity and learned helplessness [80]. Motor laterality is equally distributed in feral horses, while in domestic horses, age [20], breed [21], training [50] and carrying a rider [61] may cause left leg preferences. Most horses initially observe novel persons and objects and potentially threatening situations with their left sensory organs, and a left shift both in sensory and motor laterality indicates that the horse experiences increased emotionality or stress [16,24,25,67]. Long-term insufficiencies in welfare, housing or training may result in left shifts in motor and sensory laterality and pessimistic mentalities [22]. Therefore, there should be a rethinking of current training methods aimed at straightening the horse [10] and emphasis placed on balance rather than straightness [61]. If the horse is truly balanced and moves its hindquarters under its balance point, it will be straight, but if it is simply straight, it is not necessarily truly balanced. There will always be a degree of morphological asymmetry as well as motor laterality, but these can be minimised with correct training and muscle development. Considering the goal of a relaxed and responsive horse, training that focuses on the longitudinal balance as well as the lateral balance should be applied [48] while accepting the horse’s natural laterality.

## Data Availability

All data used for the study are included in the manuscript.

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
