# Peer review of "Laterality in Horse Training: Psychological and Physical Balance and Coordination and Strength Rather Than Straightness"

_animals, 2022, doi:10.3390/ani12081042_

Round 1

Reviewer 1 Report

Dear Authors, please correct the marked word "mental", because this is for humans and not for animals. The mentality is described in personality, psychology and sociology in humans and therefore not in animals! Please change it with "behavioural". This is the right term for animals.

Author Response

Review 1

Dear Authors, please correct the marked word "mental", because this is for humans and not for animals. The mentality is described in personality, psychology and sociology in humans and therefore not in animals! Please change it with "behavioural". This is the right term for animals.

  • Dear referee, thank you very much for your suggestion. We clarifyed the terminology “mental” wherever possible. In most places, with the term “mental” we describe the brain activity. Behaviours are just the expression of brain activity. “Behaviour” under any interpretation means the outward presentation of emotion or mental processes, so is not a suitable replacement for “mental”, or “psychological” for that matter.

Therefore, we exchanged the wording “mental” with “psychological” in most places. The term “psychological” is more commonly applied in animals when describing brain activity than “mental”. Only in the parts where we talk about laterality and mentality in horses it is not scientifically correct to remove the term “mental” as the state of the brain when horses express certain mentalities, such as pessimism and optimism, needs to be termed as its “mental state”.

  • We agree that: “during gestation period” would be more suitable for line 113 and changed the wording.

Reviewer 2 Report

Comments

This review is comprehensive on the function and causes of motor and sensory laterality in horses. It is also very constructive, readable and with a bit new approach of how to see laterality and asymmetry in the horse, and of what elements (balance, coordination and equal strength on both sides) should be emphasized and focused on when dealing with it in training.

The message the reviewer takes from this review is that the authors want to emphasize asymmetry/laterality as a natural phenomenon in the horse which should be more readily accepted by trainers and riders. But that the training-methods to achieve “straightness” should not focus on that word (but rather on balance, coordination and equal strength on both sides), be constructive, emphasize welfare and never be forcing. However, at the same time the authors accept the knowledge that straightness of the horse used for riding is important for quality of gaits, performance, health and longevity.

Also the authors have the new approach that laterality and how it develops (decreases/increases) with training (housing etc.) might be an important indicator concerning welfare of the horse.

Other comments:

Line 290. Wells und Blanche, should be Wells and Blanche.

Line297. Same as in line 290. Change und to and.

Line 401. Cardiac rate.  Change to heart rate. That is more clear and in line with the reference behind it.    Change because:  Cardiac output = stroke volume x heart rate. Part of the cardiac activity is heart rate.

Author Response

Review 2

Comments

This review is comprehensive on the function and causes of motor and sensory laterality in horses. It is also very constructive, readable and with a bit new approach of how to see laterality and asymmetry in the horse, and of what elements (balance, coordination and equal strength on both sides) should be emphasized and focused on when dealing with it in training.

The message the reviewer takes from this review is that the authors want to emphasize asymmetry/laterality as a natural phenomenon in the horse which should be more readily accepted by trainers and riders. But that the training-methods to achieve “straightness” should not focus on that word (but rather on balance, coordination and equal strength on both sides), be constructive, emphasize welfare and never be forcing. However, at the same time the authors accept the knowledge that straightness of the horse used for riding is important for quality of gaits, performance, health and longevity.

Also the authors have the new approach that laterality and how it develops (decreases/increases) with training (housing etc.) might be an important indicator concerning welfare of the horse.

  • Dear referee, thank you very much for your supporting lines. We are very much delighted that you found the literature review on laterality in horses useful and that the intention and message of our manuscript was understood so well.

Other comments:

Line 290. Wells und Blanche, should be Wells and Blanche.

  • done

Line297. Same as in line 290. Change und to and.

  • done

Line 401. Cardiac rate.  Change to heart rate. That is more clear and in line with the reference behind it.    Change because:  Cardiac output = stroke volume x heart rate. Part of the cardiac activity is heart rate.

  • We agree, has been changed

Reviewer 3 Report

 This is a thorough review of laterality in horses.   The hypothesis that  training  to reduce laterality is a stress was not  upheld by data.

81  should mention sex differences in laterality  Also  isn't there a contrast in populations Some becoming more lateralized with age and others not.

90 what are electric markers on a treadmill?

112 In humans handedness seems to be genetic not a result of development.

131  mostly neurologically What are the  rest of the connections Do you mean connected neurologically mainly to the contralateral side.

152 absolving  I don't think you mean forgiving sins. Perhaps you mean performing

279 define cut a circle

308 does this advantage change when the horse run in a different direction- clockwise vs counter clockwise

 337 might reiterate that the age and training confounded that study

388 not sure what this method was. The horse approaches the bucket 20 times and then foreleg preference was measured.

Author Response

Review 3

This is a thorough review of laterality in horses.   The hypothesis that training  to reduce laterality is a stress was not  upheld by data.

  • Dear referee, thank you very much for investing your time and for your thoughtful comments. We rewrote the abstract and the conclusion in the light of your and the fourth editors suggestions.

81  should mention sex differences in laterality  Also  isn't there a contrast in populations Some becoming more lateralized with age and others not.

  • In the parts further below, we discussed effects on laterality of all the factors we found in the literature in detail. At this point we intended to give a short definition on the different forms of laterality, as they are confused in the literature quite often. As other referees found the manuscript easy to read and to understand we feel that the structure of the manuscript is helpful for understanding the complex topic of laterality and would like to keep the present order of the content.

90 what are electric markers on a treadmill?

  • Yes, you are right, this point was unclear. We now wrote: “…measurements taken with electric markers attached to horses body when walking and trotting on a treadmill.”
  • humans handedness seems to be genetic not a result of development.
  • This line refers to body asymmetry, not to handedness. Handedness is described further below. The interaction of genetic and developed handedness is discussed in great detail. Humans do not differ from other mammals in this concern.
  • Partially handedness is genetic and innate, partially it is developed in early childhood. However, handedness manifests when mammals and humans mature. After the manifestation of handedness changing handedness is considered to cause psychological imbalance in humans.

131  mostly neurologically What are the  rest of the connections Do you mean connected neurologically mainly to the contralateral side.

  • Thank you, your point is well taken. The word “mostly” was misplaced. It should say: The left and right hemispheres are neurologically connected mostly to the contralateral side of the body. Has been changed.

152 absolving  I don't think you mean forgiving sins. Perhaps you mean performing

  • Correct! This is a funny German – English translation error J. Has been corrected.

279 define cut a circle

  • Fine, has been done.

308 does this advantage change when the horse run in a different direction- clockwise vs counter clockwise

  • We agree, this point needs further clarification. We added: “Thourougbreds and Arabians showed this preference when running around an oval racetrack in anticlockwise direction and Quarter horses on a straight line.”

Reviewer 4 Report

Thank you for the submission of an excellent quality manuscript. It was easy to read and clarified aspects of laterality in horses. The unique angle of looking at laterality in regards to training and welfare was appreciated.

I have only a few minor grammatical corrections as outlined below, and a couple of comments for consideration.

L46 - the keywords of straightness and training are not needed as they appear already in the title

L113 - change "before gestation" to read "during gestation"

L152 - the word "absolving" does not seem to be the correct word choice. maybe "undertaking"? 

L262 - missing a word in this sentence. should read "...strong uneveness in the feet show strong motor laterality..."

L273 - on the topic of motor laterality - note that thoroughbreds did not change their laterality based on their training as there was no difference between those racing clockwise vs counterclockwise. there might seem to be an advantage in switching the leading lead when going around corners. see Differences in motor laterality between breeds of performance horse. Paul D.McGreevy Peter C.Thomson. https://doi.org/10.1016/j.applanim.2005.09.010

L480 - it might be useful to also consider the following paper showing thoracic asymmetry in horses and the possible link to issues with saddle fit. see Merkies K, Alebrand J, Harwood B, LaBarge K, Scott L. 2020. Investigation into thoracic asymmetry in ridden horses. Comparative Exercise Physiology, 16(1):55-62. https://doi.org/10.3920/CEP190025  

L554 - as horses are sentient beings, they should not be referred to as "it"

L602 - this conclusion seems to focus only on the last point discussed in the discussion and emphasizes as aspect that was only mildly discussed or not at all. the use of equipment and forced training practices was not explicitly discussed but appears here in the conclusion. 
I feel the conclusion should be more broad to encompass everything that was discussed and not focus on the negative aspects of horse training. 

this comment applies also to the abstracts which seem to overstate training practices and equipment as a negative effect on horse welfare when this was not explicitly presented in the paper.

Author Response

Review 4

Thank you for the submission of an excellent quality manuscript. It was easy to read and clarified aspects of laterality in horses. The unique angle of looking at laterality in regards to training and welfare was appreciated.

  • Dear referee, thank you very much for your supporting lines. We are very much delighted that you found the literature review on laterality in horses useful and that the intention and message of our manuscript was understood so well.

I have only a few minor grammatical corrections as outlined below, and a couple of comments for consideration.

L46 - the keywords of straightness and training are not needed as they appear already in the title

  • Fine, has been deleted

L113 - change "before gestation" to read "during gestation"

  • Yes you are right, wen now wrote “during gestation period”

L152 - the word "absolving" does not seem to be the correct word choice. maybe "undertaking"? 

  • Correct! This is a funny German – English translation error J. Has been changed to “performing”.

L262 - missing a word in this sentence. should read "...strong uneveness in the feet show strong motor laterality..."

  • The verb was left out because this part of the sentence refers to the main sentence where we used “displayed”. But you are right. The sentence is more easy to read when “showed” is included. We did so.

L273 - on the topic of motor laterality - note that thoroughbreds did not change their laterality based on their training as there was no difference between those racing clockwise vs counterclockwise. there might seem to be an advantage in switching the leading lead when going around corners. see Differences in motor laterality between breeds of performance horse. Paul D.McGreevy Peter C.Thomson. https://doi.org/10.1016/j.applanim.2005.09.010

  • Thank you for pointing us to these two aspects. We changed the text respectively.

L480 - it might be useful to also consider the following paper showing thoracic asymmetry in horses and the possible link to issues with saddle fit. see Merkies K, Alebrand J, Harwood B, LaBarge K, Scott L. 2020. Investigation into thoracic asymmetry in ridden horses. Comparative Exercise Physiology, 16(1):55-62. https://doi.org/10.3920/CEP190025  

  • Yes indeed, very interesting. We integrated the manuscript in the text and the discussion.

L554 - as horses are sentient beings, they should not be referred to as "it"

  • Fine, has been reworded.

L602 - this conclusion seems to focus only on the last point discussed in the discussion and emphasizes as aspect that was only mildly discussed or not at all. the use of equipment and forced training practices was not explicitly discussed but appears here in the conclusion. 
I feel the conclusion should be more broad to encompass everything that was discussed and not focus on the negative aspects of horse training. 

this comment applies also to the abstracts which seem to overstate training practices and equipment as a negative effect on horse welfare when this was not explicitly presented in the paper.

  • We agree. The abstract and the conclusion has been rewritten. We hope they provide suitable information about the content of the manuscript now.
